# Are total omega-3 and omega-6 polyunsaturated fatty acids predictors of fatal stroke in the Adventist Health Study 2 prospective cohort?

**Alan Cupino** [ID]*, **Gary Fraser, Synnøve Knutsen, Raymond Knutsen, Celine Heskey, Joan Sabaté, David Shavlik**

School of Public Health, Loma Linda University, Loma Linda, CA, United States of America

* acupino@llu.edu

## Abstract

### Introduction

The effects of omega-3 (n-3) and omega-6 (n-6) polyunsaturated fatty acids (PUFA) on cerebrovascular disease remain unsettled. However, most studies have focused on marine sourced n-3 PUFA rather than total n-3 PUFA, of which the majority in the American diet is plant derived. This study therefore intended to investigate these effects in a cohort for which the vegetarian diet was more prevalent than the general public.

### Methods

Cox proportional hazards with fatal stroke as the outcome was performed on the approximately 96,000 subject Adventist Health Study 2 prospective cohort. Stratification by race and sex was performed on models with a priori covariables, comparing 90th to 10th percentile daily intakes of energy-adjusted total n-3 PUFA, total n-6 PUFA, and the n-6 / n-3 PUFA ratio as variables of interest.

### Results

For the main analytical group (78,335 subjects), the hazard ratio (95% confidence interval) for total n-3 PUFA was 0.65 (0.51–0.83), and for total n-6 PUFA was 1.37 (1.02–1.82), while adjusting for both fatty acids in the model. The n-6 / n-3 PUFA ratio was harmful with a HR of 1.40 (1.16–1.69), whereas the inclusion of total n-3 PUFA slightly attenuated the HR to 1.33 (1.02–1.74). Effects were similar for the non-black sex-combined and sex-specific analyses.

### Conclusion

In most analytic groups, subjects with greater total n-3 PUFA intakes have lower risk of fatal stroke, and those with a higher n-6 / n-3 PUFA ratio had higher risk. However, the n-6 / n-3 PUFA ratio remains statistically significant even after adjusting for total n-3 PUFA or total n-

**Data Availability Statement:** The authors' agreement with the Adventist Health Study does not allow publishing data used for this manuscript

or sharing it with others. The AHS-2 Coordinators' Committee would cooperate in case any fraud or forgery is suspected in manuscripts which use AHS-2 data. If researchers would like access to data underlying the results presented in the study, please contact the AHS-2 at https://AdventistHealthStudy.org/about/contact-us.

**Funding:** The authors received no specific funding for this work.

**Competing interests:** The authors have declared that no competing interests exist.

6 PUFA, suggesting that the ratio is of epidemiologic interest for cerebrovascular disease research.

## Introduction

Cerebrovascular diseases, the majority of which are strokes, make up the fifth most common cause of death in the United States [1]. Stroke mortality rates have recently started to decline, in tandem with falling hypertension rates [2–4]; however, risk factors for stroke remain highly prevalent in the US population, leading to a burden on the American healthcare system [5, 6]. Two dietary fats which may affect the pathologic process resulting in stroke have received attention recently for their potential mental, neurological, and cardiovascular health effects: omega-3 (n-3) and omega-6 (n-6) polyunsaturated fatty acids (PUFA) [7].

Many studies have investigated n-3 PUFA in relationship to cardiovascular disease; however, most have examined levels of marine-sourced long-chain PUFA such as eicosapentaenoic acid (EPA) and docosahexaenoic acid (DHA), or daily fish intake as a proxy. The vegetarian-sourced a-linolenic acid (ALA), which Americans eat more than 8 times the combined amount of EPA and DHA, is poorly described despite being an essential fatty acid [8, 9].

Few studies have focused on fatal stroke as an endpoint [10–20], and within the handful of fatal stroke cohort reports and clinical trials, the observed effects are typically not statistically significant. The Singapore Chinese Health Study reported a protective hazard ratio comparing quartiles of daily total n-3 PUFA intake, though did not quite reach statistical significance [13]. The Japan Collaborative Cohort Study for Evaluation of Cancer (JACC) also reported a non-statistically significant protective hazard ratio comparing total n-3 PUFA quintiles [18]. Similarly, dietary interventions like the 46,558 subject Women's Health Initiative Diet Modification Trial from the United States have failed to produce any statistically significant results [12, 16]. Despite most individual studies failing to reach statistical significance, the results of meta-analyses are mixed. A 2015 meta-analysis of prospective cohorts looking at long chain fatty acids and fatal stroke had a significant protective effect, whereas a 2020 meta-analysis of randomized clinical trials found no such statistically significant results [21, 22].

While n-3 PUFA appears to be either protective or have no association with fatal stroke, n-6 PUFA are not observed to be associated with stroke in either direction [17, 23]. The Cardiovascular Health Study from the United States reported a slightly harmful ischemic stroke hazard ratio comparing interquintile arachidonic acid, an n-6 PUFA, as percentage of total fatty acid in plasma [17]. The Kuopio Ischaemic Heart Disease Risk Factor Study (KIHD) from Finland reported a slightly protective ischemic stroke hazard ratio comparing quartile 4 to quartile 1 serum n-6 PUFA [24]. Neither study achieved statistical significance.

The literature therefore remains mixed on whether n-3 PUFA protect against fatal stroke, and whether n-6 PUFA are protective or harmful [19, 20]. It is worth noting that most current PUFA studies either did not have sufficient power for their hypotheses [25], did not study total n-3 or n-6 PUFA [23], did not adjust for the effect of the other PUFA type (n-3 or n-6) [26], or did not have fatal stroke as an outcome of interest.

Furthermore, prior studies did not adjust the n-6 / n-3 PUFA ratio by total n-3 PUFA amounts, a shortcoming of investigating the n-6 / n-3 PUFA ratio which has been a topic of previous contention in the field of PUFA research [23, 27, 28]. An n-6 / n-3 PUFA ratio of 5:1 would describe an intake of 50 mg n-6 PUFA and 10 mg n-3 PUFA as identical to an intake of 5 mg n-6 PUFA and 1 mg n-3 PUFA, despite strikingly different n-6 PUFA (50 mg to 5 mg)

and n-3 PUFA (10 mg to 1 mg) raw intakes. Therefore, adjusting for either the numerator or denominator would be helpful to differentiate identical ratios taken from very different raw intakes, even after adjusting for total energy intake.

The AHS-2 cohort described herein has a greater total n-3 PUFA intake overall than other fatal stroke cohort studies. This is likely due to the abundance of ALA in the vegetarian diet as a proportion of total n-3 PUFA intake relative to the minimal quantity of EPA and DHA from marine life in the typical Western diet [9, 29, 30]. Using data from a validated food frequency questionnaire (FFQ), this study sought to investigate the effects of n-3 PUFA, n-6 PUFA, and the ratio between the two on the risk of stroke mortality in a large prospective cohort [31]. Unique to our study, we also investigated the impact of the ratio adjusting for the absolute amount of n-3 or n-6 PUFA.

## Methods

Risk of fatal stroke was assessed in the Adventist Health Study-2 (AHS-2) prospective cohort according to differences in polyunsaturated fatty-acid (PUFA) intakes as the exposure [32]. The AHS-2 has been described in detail previously by Butler et. al, but in summary is a prospective cohort study of approximately 96,000 individuals from U.S. and Canada over the age of 30 at enrollment, with collection of baseline data between 2002–2007 [32]. The Loma Linda University School of Public Health Institutional Review Board approved the use of anonymized data for these analyses, with informed consent attained from each study participant in writing. Of the total participants, 26% racially self-identified as black [33]. Subjects had limited or no current exposure to tobacco and alcohol, so variables controlling for those factors were collapsed [32]. The AHS-2 dataset includes demographics, medical history, and vital status from linkages with the NDI (National Death Index). Nutrient intakes, including supplementation, were calculated from the validated food frequency questionnaire (FFQ) data [31], using Nutrition Data System for Research software (2008, Nutrition Coordinating Center, University of Minnesota, Minneapolis, MN). The FFQ requested that subjects report dietary intakes considering the previous twelve months.

Subjects reporting a previous stroke or myocardial infarction at baseline were excluded from analysis due to the increased likelihood of subsequent strokes and other potential biases including reverse causation. Fatal stroke was determined by ICD codes I61-64 as any cause of death in the NDI data. NDI data on stroke deaths is non-specific (greater than 80% of the reported stroke mortalities are coded I64 "stroke, not specified as haemorrhage or infarction") [34]. Thus, we were unable to include type of stroke despite acknowledging the etiologic differences [35].

Exclusion criteria were applied with stepwise removal, starting at 95,597 subjects in the AHS-2 with available data: 5,296 subjects had invalid dietary data or insufficient, extreme, or un-imputable missing data such as birthdate or body mass index (BMI). Participants with the following prevalent factors were excluded: 7,344 reported prevalent cancer (excluding non-melanoma skin cancer), at the beginning of the study; 3,097 reported a previous myocardial infarction, congestive heart failure, coronary stent, coronary artery bypass, or surgery on their carotid arteries; 671 reported a previous stroke. Lastly, 854 subjects had values outside the range of 14–59 kg/m$^2$ for BMI, greater than 7 feet for height, less than 500 or greater than 4,500 kcal daily caloric intake, leaving 78,335 analytic subjects. Within this final analytical group, 37% were either vegan or lacto-ovo vegetarian, and 16% were either pesco-vegetarian or semi-vegetarian. Sub-group analysis stratified by race was also performed, where the black sub-group included 20,596 subjects, and the non-black sub-group contained the remaining 57,739 subjects.

## Study variables

Primary exposures included total n-3 PUFA, total n-6 PUFA, and the ratio of n-6 to n-3 PUFA. All three variables were treated as continuous and reflected the combined intake of diet and supplementation. Supplemented nutrients included saturated fatty acids, mono-unsaturated fatty acids, poly-unsaturated fatty acids, fiber, sodium, vitamin C, alpha-tocopherol equivalents, and flavonoids. Approximately 12% of participants supplemented n-3 PUFA, and approximately 15% supplemented n-6 PUFA. However, overall rates of supplementation were low: the 90th percentile of n-3 PUFA supplement contribution was 8.5% of total n-3 PUFA intake, and the corresponding 90th percentile of total n-6 PUFA intake due to n-6 PUFA supplements was 0.1%. Validity correlation between FFQ and 24 h dietary recalls were 0.53 (95% CI = 0.47–0.59) for whites, 0.43 (0.35–0.51) for blacks [31]. These exposures were analyzed both individually and in two specific combinations: n-3 PUFA with n-6 PUFA, and the n-6 / n-3 PUFA ratio with n-3 or n-6 PUFA. All three primary exposures were log transformed and energy adjusted using the residual method [36]. Each variable compared the 90th to the 10th percentile of the residual [36].

Variables were selected based on a-priori knowledge and decisions. Modifiable factors including diet, infrequent physical activity, smoking, and alcohol have all been associated with risk of stroke mortality, as well as increased age and the male sex, so adjustment for these factors was necessary [37–44].

Seven categorical covariates and one calculated variable, comprising 18 total levels, were covariates included in the primary analysis: hypertension diagnosis; diabetes diagnosis; education level (3 levels); use of cholesterol-lowering medication; use of aspirin (3 levels); current and/or prior use of tobacco; current and/or prior use of alcohol; and a variable which combined sex, menopausal status, and hormone replacement therapy (4 levels).

Eleven continuous covariates were included in the full primary analysis: BMI; minutes of exercise per week; saturated fatty acids; monounsaturated fatty acids; glycemic index, with glucose as the reference; total dietary fiber; dietary salt; vitamin C; total alpha-tocopherol equivalents; and a flavonoid variable created by summing all available flavonoid variables including supplemental flavonoids and isoflavones; and total energy [36].

## Statistical analyses

Surviving subjects were right censored on December 31, 2015, to coincide with the final NDI linkage, and left censored at the age they joined AHS-2. Multiple imputation was performed for variables with missing data, and select dietary data underwent guided imputation [45]. Descriptive statistics that included means and standard deviations for continuous variables, and counts and percentages for categorical variables, were calculated for all variables stratified by median intake of n-3 and n-6 PUFA. Additionally, statistical tests were conducted between the below-median and above-median groups using ANOVA for continuous variables and chi-square for categorical.

Multivariable Cox proportional hazard regressions with attained-age as the time variable were performed using Statistical Analysis Software (9.4, SAS Institute Inc, Cary, NC), with hazard ratios comparing the 90th to the 10th percentile of the residuals of the main exposures. Non-linearity was assessed using Martingale residuals, and the proportional hazards assumption was checked by Schoenfeld residuals resulting in the inclusion of a time-interaction variable for diabetes in the model to account for non-proportional levels. Rubin's Rules were used post-imputation to pool estimates for interpretation [46].

After adjusting for covariates, four models were investigated: model 1 examined either total n-3 PUFA, total n-6 PUFA, or the n-6 / n-3 PUFA ratio in isolation; model 2 included n-3

PUFA alongside n-6 PUFA; model 3 included n-3 PUFA and the n-6 / n-3 PUFA ratio; and model 4 included n-6 PUFA and the n-6 / n-3 PUFA ratio. For model 2, both PUFA variables were included to determine the effect of adjusting for their direct competitor for binding sites. For the n-6 / n-3 PUFA ratio in models 3 and 4, n-3 or n-6 PUFA were included to serve as an index for comparing identical ratios with different absolute amounts. Sub-group analysis was also performed, stratifying first by race, then by sex.

## Results

When creating high- and low-intake groups by stratifying across the median daily intake of n-3 and n-6 PUFA, the mean n-6 / n-3 PUFA ratio was lower for both the high-intake n-3 PUFA (7.9:1 compared to 10.1:1) and low-intake n-6 PUFA groups (9.6:1 compared to 8.4:1). All means and category frequencies between low- and high-intake groups for both n-3 and n- 6 PUFA were different, with the exception of minutes of exercise per week, statin use in the n-6 PUFA comparison, and smoking status in the n-3 PUFA comparison (Table 1).

Over an average follow-up of 11.2 years, there were 715 fatal strokes among 78,335 subjects (proportional incidence 0.91%), 455 of which were in females and 260 in males. In the sub-group analysis, there were 123 (0.6%) fatal strokes in the black group, and 592 (1.03%) fatal strokes in the non-black group. Median age at study enrollment for the non-black group was 6 years older than the black group.

Regression models produced hazard ratios for fatal stroke comparing the $90^{th}$ to the $10^{th}$ percentile of the variable of interest (Table 2). The energy-adjusted $10^{th}$ percentile daily values for n-3 PUFA, n-6 PUFA, and the n-6 / n-3 PUFA ratio were 2.38 g / day, 15.58 g / day, and 6.11:1, and the $90^{th}$ percentiles were 3.25 g / day, 16.45 g / day, and 12.07:1.

Higher n-3 PUFA intake alone was associated with lower risk of fatal stroke with a hazard ratio (HR) of 0.74 (0.60–0.92) (Table 2). Adjusting for n-6 PUFA strengthened the protective association of n-3 PUFA to 0.65 (0.51–0.83). When n-3 PUFA and the n-6 / n-3 PUFA ratio are together in the same model, the effect of n-3 PUFA largely disappears. The effect of n-3 PUFA was stronger in men than in women, and when adjusting for n-6 PUFA was still significantly protective in each sex, with HRs of 0.58 (0.38–0.88) for males and 0.69 (0.51–0.94) for females. Only in the sex-combined model alongside n-3 PUFA did n-6 PUFA show a statistically significant increase in hazard of fatal stroke with a HR of 1.37 (1.02–1.82), although the sex specific results were in the same direction. The n-6 / n-3 PUFA ratio was associated with an increased risk of fatal stroke with a HR of 1.40 (1.16–1.69), and adjusting for n-3 PUFA weakened the effect somewhat down to a HR of 1.33 (1.02–1.74). For the sex specific results, the n-6 / n-3 PUFA ratios alone were significant for both sexes but were not significant when n-3 PUFA was added to the model.

When stratifying by race (Table 3), no statistically significant effects were observed for the black analytical group, for whom most point estimates were in the opposite direction from the combined and non-black analytical groups. Due to the low number of events relative to the number of variables, analysis stratified by sex was not performed in the black sub-group. For all analyses of non-black participants, n-3 PUFA was statistically significant when alone, with a HR of 0.68 (0.54–0.85) in model 1, and while adjusting for n-6 PUFA in model 2 became even more protective with a HR of 0.55 (0.42–0.72). Again, the possible harmful effect of n-6 PUFA was amplified when adjusting for n-3 PUFA in model 2 (HR = 1.60; 1.15–2.22) compared to n-6 PUFA alone in model 1 for all non-blacks (HR = 1.07; 0.82–1.41), with similar effect for males (HR = 1.86; 1.05–3.30). The non-black female hazard ratio of 1.48 (0.95–2.29) was close to statistical significance. For the n-6 / n-3 PUFA ratio, all non-black groups saw a statistically significant harmful effect, except for the female model 3 at 1.43 (0.96–2.15) which also

**Table 1. Mean and standard deviation, or count and proportion, for demographic variables comparing those lower and higher than the medians of total n-3 PUFA or total n-6 PUFA (Chi-Squared and ANOVA p-values for comparison within fatty acid type).**

| | Mean (StdDev) | | | | | | |
| --- | --- | --- | --- | --- | --- | --- | --- |
| | Total n-3 PUFA | | | | Total n-6 PUFA | | |
| | Low | High | p | | Low | High | p |
| **Age** (y) | 56.9 (14.1) | 57.1 (13.6) | 0.045 | | 56.3 (13.9) | 57.7 (13.8) | <0.001 |
| **n-6 / n-3 ratio** | 10.1:1 (2.9) | 7.9:1 (2.0) | <0.001 | | 8.4:1 (2.4) | 9.6:1 (2.9) | <0.001 |
| **BMI** (kg/m^2) | 26.9 (5.9) | 27.3 (5.9) | <0.001 | | 27.2 (5.9) | 27.0 (5.8) | <0.001 |
| **Kilocalories** | 1912.9 (759.9) | 1923.9 (761.0) | 0.043 | | 1924.7 (773.8) | 1912.0 (746.9) | 0.020 |
| **Exercise** (min/wk) | 80.8 (96.9) | 81.7 (95.2) | 0.169 | | 81.5 (97.4) | 81.0 (94.7) | 0.516 |
| **Saturated Fat** (g/day) | 16.3 (9.7) | 18.0 (9.8) | <0.001 | | 16.5 (10.0) | 17.9 (9.5) | <0.001 |
| **Mono-unsaturated Fat** (g/day) | 26.3 (14.9) | 31.4 (16.6) | <0.001 | | 24.7 (13.5) | 32.9 (17.2) | <0.001 |
| **Glycemic Load** | 134.8 (58.5) | 118.9 (51.1) | <0.001 | | 137.5 (59.9) | 116.2 (48.5) | <0.001 |
| **Fiber** (g/day) | 34.7 (17.5) | 33.5 (16.7) | <0.001 | | 35.1 (18.2) | 33.2 (15.8) | <0.001 |
| **Sodium** (g/day) | 3.3 (1.9) | 3.5 (2.1) | <0.001 | | 3.2 (1.9) | 3.6 (2.1) | <0.001 |
| **Vitamin C** (mg/day) | 482.9 (524.2) | 571.2 (606.3) | <0.001 | | 517.8 (557.7) | 536.2 (578.8) | <0.001 |
| **Total Alpha-tocopherol Equivalents** (mg/day) | 131.4 (202.0) | 170.1 (236.3) | <0.001 | | 138.4 (213.3) | 163.1 (227.1) | <0.001 |
| **Total Isoflavones** (mg/day) | 13.3 (18.1) | 23.3 (29.7) | <0.001 | | 13.1 (19.0) | 23.5 (29.1) | <0.001 |
| | Count (% Variable) | | | | | | |
| | Total n-3 PUFA | | | | Total n-6 PUFA | | |
| | Low | High | | | Low | High | |
| | [39,168] | [39,167] | p | | [39,167] | [39,168] | p |
| **Hypertension** - *Ever Diagnosed* | 10484 (26.8) | 11663 (29.8) | <0.001 | | 11278 (28.8) | 10869 (27.8) | 0.001 |
| **Diabetes** - *Ever Diagnosed* | 2750 (7.0) | 3475 (8.9) | <0.001 | | 2908 (7.4) | 3317 (8.5) | <0.001 |
| **Combined Sex** | | | <0.001 | | | | <0.001 |
| *Male* | 14972 (38.6) | 12599 (32.5) | | | 13910 (35.9) | 13661 (35.2) | |
| *Female, Pre-menopause* | 7689 (19.8) | 7946 (20.5) | | | 8147 (21.0) | 7488 (19.3) | |
| *Female, Post-Meno., No HRT* | 7456 (19.2) | 7954 (20.5) | | | 7934 (20.5) | 7476 (19.2) | |
| *Female, Post-Meno., w/ HRT* | 8713 (22.4) | 10273 (26.5) | | | 8748 (22.6) | 10238 (26.3) | |
| **Race** - *Black* | 9518 (24.3) | 11078 (28.3) | <0.001 | | 11807 (30.1) | 8789 (22.4) | <0.001 |
| **Education** | | | <0.001 | | | | <0.001 |
| *High School and Below* | 8498 (21.7) | 7766 (19.8) | | | 8823 (22.5) | 7441 (19.0) | |
| *Some College* | 15226 (38.9) | 15814 (40.4) | | | 15391 (39.3) | 15649 (40.0) | |
| *Bachelors and Above* | 15443 (39.4) | 15588 (39.8) | | | 14955 (38.2) | 16076 (41.1) | |
| **Statin** - *Ever Used* | 3567 (9.1) | 3914 (10.0) | <0.001 | | 3741 (9.6) | 3740 (9.6) | 0.993 |
| **Aspirin 05** | | | <0.001 | | | | 0.126 |
| *Never use* | 28541 (72.9) | 27729 (70.8) | | | 28204 (72.0) | 28066 (71.7) | |
| *< Daily* | 5484 (14.0) | 6025 (15.4) | | | 5827 (14.9) | 5682 (14.5) | |
| *>= Daily* | 5142 (13.1) | 5414 (13.8) | | | 5138 (13.1) | 5418 (13.8) | |
| **Smoking** - *Ever Used* | 7636 (19.35) | 7579 (19.4) | 0.606 | | 7974 (20.4) | 7241 (18.5) | <0.001 |
| **Drinking** - *Ever Used* | 15953 (40.7) | 16781 (42.8) | <0.001 | | 17213 (44.0) | 15521 (39.6) | <0.001 |

approached statistical significance. With n-3 PUFA also in the model, the n-6 / n-3 PUFA ratio for all non-blacks had a HR of 1.54 (1.14–2.08) and for males a HR of 1.77 (1.05–3.00).

## Discussion

The present study demonstrates a protective association for total n-3 PUFA, a possible harmful association for total n-6 PUFA intake when adjusting for total n-3 PUFA intake, and a harmful

**Table 2. Adjusted hazard ratios for total n-3 PUFA, total n-6 PUFA, and the n-6 / n-3 PUFA ratio.**

| | HR (95%CI) - 90th vs 10th percentile daily intake | | |
| --- | --- | --- | --- |
| | **Total n-3 PUFA** | **Total n-6 PUFA** | **n-6 / n-3 PUFA ratio** |
| | [3.25 vs. 2.38 g/d][#] | [16.45 vs. 15.58 g/d][#] | [12.07:1 vs. 6.11:1][#] |
| **Sex-combined (715 events)** | | | |
| Model 1 (individual) | 0.74 (0.60–0.92)** | 1.04 (0.81–1.33) | 1.40 (1.16–1.69)*** |
| Model 2 | 0.65 (0.51–0.83)*** | 1.37 (1.02–1.82)* | |
| Model 3 | 0.93 (0.69–1.25) | | 1.33 (1.02–1.74)* |
| Model 4 | | 0.94 (0.72–1.21) | 1.42 (1.17–1.72)*** |
| **Female (455 events)** | | | |
| Model 1 (individual) | 0.77 (0.59–0.99)* | 1.00 (0.72–1.38) | 1.33 (1.04–1.69)* |
| Model 2 | 0.69 (0.51–0.94)* | 1.26 (0.86–1.85) | |
| Model 3 | 0.90 (0.61–1.33) | | 1.24 (0.87–1.76) |
| Model 4 | | 0.91 (0.65–1.28) | 1.35 (1.05–1.72)* |
| **Male (260 events)** | | | |
| Model 1 (individual) | 0.70 (0.50–0.99)* | 1.10 (0.73–1.66) | 1.54 (1.11–2.13)** |
| Model 2 | 0.58 (0.38–0.88)* | 1.55 (0.95–2.54) | |
| Model 3 | 0.96 (0.59–1.56) | | 1.50 (0.95–2.36) |
| Model 4 | | 0.96 (0.63–1.47) | 1.55 (1.11–2.16)* |

#: 90th percentile vs. 10th percentile daily intake

Model 1: total n-3 PUFA alone, or total n-6 PUFA alone, or the n-6/n-3 PUFA ratio alone

Model 2: Total n-3 and n-6 PUFA

Model 3: Total n-3 PUFA and n-6 / n-3 PUFA ratio

Model 4: Total n-6 PUFA and n-6 / n-3 PUFA ratio

Covariates:

Male—age, race, bmi, kilocalories, minutes of exercise, saturated fat, fiber, mono-unsaturated fat, glycemic load, sodium, vitamin C, total alpha-tocopherol equivalents, total isoflavones, hypertension, diabetes, education level, statin use, aspirin, tobacco use, alcohol use

Female—[Male covariates], menopause status, hormone-replacement therapy

Sex-combined—[Female covariates], sex

*p < .05

**p < .01

***p < .001

association for a higher n-6 / n-3 PUFA ratio. Results were similar between men and women, controlling for age and additional dietary and lifestyle behaviors that are associated with risk of stroke. Additionally, the median intake was higher than other studies in which fish was a frequent dietary source for n-3 PUFA and a vegetarian diet was uncommon. In contrast, the AHS-2 study examined the impact of total n-3 PUFA that was mostly from plant sources like flaxseed and walnuts, and some vegetable oils such as soy oil [47, 48]. This predominance of plant-based ALA is likely due to the high proportion of vegetarians in the study population whose intake of n-3 PUFA comes entirely from plants, and others for whom their minimal marine sourced PUFA intake was entirely from supplements. While ALA supplementation does exist in the AHS-2, 90th percentile daily intake of ALA was below 4 percent of total PUFA. It is unclear why the black analytical group's results differ from the non-black group's results. However, it is known that the two groups have different dietary patterns, with the black group having higher levels of n-3 and n-6 PUFA from animal sources and possibly different food preparation methods. Further investigation is indicated.

**Table 3. Adjusted race- and sex-specific hazard ratios for total n-3 PUFA, total n-6 PUFA, and the n-6 / n-3 PUFA ratio.**

| | HR (95%CI) - 90th vs. 10th percentile daily intake | | |
| --- | --- | --- | --- |
| | Total n-3 PUFA [3.25 vs. 2.38 mg/d]# | Total n-6 PUFA [16.45 vs. 15.58 mg/d]# | n-6 / n-3 PUFA ratio [12.07:1 vs. 6.11:1]# |
| **Sex-combined black (123 events)** | | | |
| Model 1 (individual) | 1.07 (0.64–1.78) | 0.86 (0.48–1.53) | 0.86 (0.57–1.30) |
| Model 2 | 1.17 (0.67–2.02) | 0.80 (0.43–1.49) | |
| Model 3 | 0.90 (0.44–1.84) | | 0.81 (0.46–1.44) |
| Model 4 | | 0.91 (0.49–1.70) | 0.88 (0.57–1.38) |
| **Sex-combined non-black (592 events)** | | | |
| Model 1 (individual) | 0.68 (0.54–0.85)*** | 1.07 (0.82–1.41) | 1.60 (1.30–1.98)*** |
| Model 2 | 0.55 (0.42–0.72)*** | 1.60 (1.15–2.22)** | |
| Model 3 | 0.94 (0.68–1.31) | | 1.54 (1.14–2.08)** |
| Model 4 | | 0.95 (0.72–1.27) | 1.61 (1.30–2.00)*** |
| **Female non-black (372 events)** | | | |
| Model 1 (individual) | 0.68 (0.51–0.90)** | 1.00 (0.70–1.45) | 1.55 (1.19–2.03)** |
| Model 2 | 0.57 (0.40–0.79)** | 1.48 (0.95–2.29) | |
| Model 3 | 0.89 (0.58–1.37) | | 1.43 (0.96–2.15) |
| Model 4 | | 0.90 (0.62–1.31) | 1.57 (1.20–2.06)*** |
| **Male non-black (220 events)** | | | |
| Model 1 (individual) | 0.68 (0.47–0.98)* | 1.20 (0.76–1.91) | 1.71 (1.19–2.46)** |
| Model 2 | 0.52 (0.33–0.81)** | 1.86 (1.05–3.30)* | |
| Model 3 | 1.05 (0.61–1.82) | | 1.77 (1.05–3.00)* |
| Model 4 | | 1.05 (0.65–1.68) | 1.70 (1.18–2.45)** |

#: 90th percentile vs. 10th percentile daily intake

Model 1: Total n-3 PUFA alone, or total n-6 PUFA alone, or the n-6 / n-3 PUFA ratio alone

Model 2: Total n-3 and n-6 PUFA

Model 3: Total n-3 PUFA and n-6 / n-3 PUFA ratio

Model 4: Total n-6 PUFA and n-6 / n-3 PUFA ratio

Covariates:

Male—age, race, bmi, kilocalories, minutes of exercise, saturated fat, fiber, mono-unsaturated fat, glycemic load, sodium, vitamin C, total alpha-tocopherol equivalents, total isoflavones, hypertension, diabetes, education level, statin use, aspirin, tobacco use, alcohol use

Female—[Male covariates], menopause status, hormone-replacement therapy

Sex-Combined—[Female covariates], sex

*$p < .05$

**$p < .01$

***$p < .001$

Since 2006, at least seven cohort studies and two clinical trials have reported on fatal stroke and n-3 PUFA intake [10–18]. Most clinical trials and cohort studies reported results suggesting a protective effect; however, results regarding stroke mortality were not statistically significant. For instance, the Japan Collaborative Cohort Study for Evaluation of Cancer Risk (JACC) Study, which reported a multivariable hazard ratio of 0.93 (0.70–1.22) for fatal stroke comparing highest and lowest quintiles of total n-3 PUFA intake. Their analytical intakes were 2.11 g / day (80th percentile) vs. 1.18 g / day (20th percentile) for 57,972 men and women, which is lower than the AHS-2's 80th percentile of 3.04 g / day and 20th percentile of 2.49 g / day [18]. The Singapore Chinese Health Study (SCHS), the only other large prospective cohort with effects that were close to statistically significant, investigated total n-3 PUFA association with fatal stroke and reported an inter-quintile risk ratio of 0.82 (0.66–1.01). Their analytical

intakes of 1.26 g / day mean total n-3 PUFA for fourth quartile (Q4) and 0.59 g / day mean for first quartile (Q1) among 60,298 men and women, are lower than the AHS-2's intakes of 3.30 g / day mean total n-3 PUFA for Q4 and 2.39 g / day for Q1 [13]. The JACC and SCHS studies controlled for n-6 PUFA but did not examine the n-6 / n-3 PUFA ratio. Importantly, neither study achieved statistical significance for the protective hazard ratios they observed with increased n-3 PUFA consumption.

Omega-6 PUFA are known to compete with n-3 PUFA in various metabolic pathways, and even interrupt n-3 PUFA uptake from plasma, which is supported by the statistically significant greater hazards for both higher n-6 PUFA and n-6 / n-3 PUFA ratios [49]. Though the n-6 / n-3 PUFA ratio in association with fatal stroke was usually statistically significant or close to significant, the majority of n-6 PUFA hazard ratios are not statistically significant. Unfortunately, due to the paucity of prospective cohort studies reporting n-6 PUFA intake in relation to fatal stroke, comparison between this study and the literature is not yet possible [50]. These results suggest an important effect that the n-6 / n-3 PUFA ratio may have on cerebrovascular mortality: any potential benefit expected by having a diet high in n-3 PUFA will likely be nullified by high levels of n-6 PUFA [51, 52]. Even adjusting for absolute n-3 PUFA intake, hazard ratios for the n-6 / n-3 PUFA ratio were only minimally affected, making the case that for both high or low n-3 PUFA consumption, the proportion of n-6 to n-3 PUFA has greater impact on the risk of stroke mortality. It should be noted that the adjusted correlation between the n-6 / n-3 PUFA ratio and n-3 PUFA was 0.65 in this study, which will have inflated variances somewhat, though the ratio remains statistically significant. While the optimal ratio has not yet been published in the National Academy of Medicine's Dietary Reference Intakes, there is growing evidence that a 1:1 ratio coincides with more beneficial, and fewer harmful, health outcomes [53]. Even so, the n-6 / n-3 PUFA ratio range in the current study population between the 10[th] percentile (6.1:1) and 90[th] percentile (12.1:1) was a large departure from the 1:1 ideal. A previous review has investigated the n-6 / n-3 PUFA ratio of various diets and noted that while the present Western diet has an n-6 / n-3 PUFA ratio of 15–20:1, other populations have much lower ratios. Examples include Japan at 4:1 and rural India at 5–6:1, and specific diets such as Swedish at 4.7:1 or Mediterranean at 2.6:1 [52].

Proposed mechanisms of protective action for n-3 PUFA include anti-inflammatory properties such as inhibition of neutrophil activity, reduced blood pressure, decreased blood viscosity, decreased triglyceride levels, anti-thrombotic effects mediated by decreased platelet aggregation, and improved insulin sensitivity [54–59]. While n-3 PUFA metabolites are generally associated with anti-inflammatory effects, n-6 PUFA metabolites are associated with pro-inflammatory effects [54, 60]. Despite this, the relationship between n-6 PUFA consumption and cardiovascular health is unclear [17, 23, 61]. However, rather than total amount of n-6 PUFA, most studies have focused on deviation from the 1:1 ratio of n-6 to n-3 PUFA, a convention supported by the present finding that the n-6 / n-3 PUFA ratio is associated with the risk of fatal stroke [61].

One strength of this study is the availability of PUFA variables derived from the validated FFQs. The median n-3 PUFA intake is also greater than other similar fatal stroke studies. As mentioned previously, the 90[th] percentile for daily energy-adjusted n-3 PUFA intake in this AHS-2 analytical sample was 3.3 g / day for females and 3.2 g / day for males, and the 10[th] percentile was 2.4 g / day for both females and males. Additional strengths of this design include guided multiple imputation, the ability to control for other dietary variables, and much lower rates of alcohol consumption and cigarette smoking relative to the national average, which limits confounding by these factors.

While this study utilized a large-scale prospective cohort, the largest study-specific limitation was a reliance on NDI data for reporting cause of death, which only distinguishes between

hemorrhagic and ischemic strokes in approximately 20-percent of cases. Additionally, validated incident stroke information is unavailable. This complicates non-fatal stroke exclusion since the criteria relied solely on subject recall and reporting. The authors recognize the difference in pathologic pathways between these two types of stroke; however, the largest proportion of total strokes are ischemic (87%), even though hemorrhagic strokes carry a greater risk of mortality [6, 62]. The proposed biological mechanisms by which polyunsaturated fatty acids improve blood vessel health, as discussed above, would directly impact both ischemic and hemorrhagic stroke outcomes. Even so, if hemorrhagic strokes are biasing the results, we presume the bias is towards the null.

In conclusion, within the AHS-2 cohort, higher n-3 PUFA intake and lower n-6 PUFA intake are protective from fatal stroke, and neither should be considered in isolation due to their direct competition in the body. The black analytic group produced a non-statistically significant opposite effect, which is a topic for future research. Finally, these results strongly support that rather than absolute intakes of n-3 or n-6 PUFA, the n-6 / n-3 PUFA ratio is important for cerebrovascular risk assessment, whether or not one concurrently adjusts for total n-3 or n-6 PUFA as well.

## Author Contributions

**Conceptualization:** Alan Cupino, Synnøve Knutsen, Raymond Knutsen, Celine Heskey, David Shavlik.

**Data curation:** Alan Cupino, David Shavlik.

**Formal analysis:** Alan Cupino, David Shavlik.

**Funding acquisition:** Alan Cupino, Gary Fraser, Synnøve Knutsen, Joan Sabaté, David Shavlik.

**Investigation:** Alan Cupino, Gary Fraser, Synnøve Knutsen, Raymond Knutsen, David Shavlik.

**Methodology:** Alan Cupino, Synnøve Knutsen, Celine Heskey, David Shavlik.

**Project administration:** Alan Cupino, Synnøve Knutsen, Raymond Knutsen, David Shavlik.

**Resources:** Alan Cupino, Gary Fraser, Synnøve Knutsen, Raymond Knutsen, David Shavlik.

**Software:** Alan Cupino, David Shavlik.

**Supervision:** Synnøve Knutsen, Raymond Knutsen, Celine Heskey, Joan Sabaté, David Shavlik.

**Validation:** Alan Cupino, Gary Fraser, Synnøve Knutsen, Celine Heskey, David Shavlik.

**Visualization:** Alan Cupino.

**Writing – original draft:** Alan Cupino.

**Writing – review & editing:** Alan Cupino, Gary Fraser, Synnøve Knutsen, Raymond Knutsen, Celine Heskey, Joan Sabaté, David Shavlik.

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
