## [Decision Letter · Decision Letter 0]

20 Jun 2022

PONE-D-22-10872Are total omega-3 and omega-6 polyunsaturated fatty acids predictors of fatal stroke in the Adventist Health Study 2 prospective cohort?PLOS ONE

Dear Dr. Cupino,

Thank you for submitting your manuscript to PLOS ONE. After careful consideration, we feel that it has merit but does not fully meet PLOS ONE’s publication criteria as it currently stands. Therefore, we invite you to submit a revised version of the manuscript that addresses the points raised during the review process.

Please submit your revised manuscript by Aug 04 2022 11:59PM. If you will need more time than this to complete your revisions, please reply to this message or contact the journal office at plosone@plos.org. Please include the following items when submitting your revised manuscript:A rebuttal letter that responds to each point raised by the academic editor and reviewer(s). You should upload this letter as a separate file labeled 'Response to Reviewers'.A marked-up copy of your manuscript that highlights changes made to the original version. You should upload this as a separate file labeled 'Revised Manuscript with Track Changes'.An unmarked version of your revised paper without tracked changes. You should upload this as a separate file labeled 'Manuscript'.

We look forward to receiving your revised manuscript.

Kind regards,

Alvaro Reischak-Oliveira, Ph.D.

Academic Editor

PLOS ONE

Journal Requirements:

Reviewers' comments:

Reviewer's Responses to Questions

**Comments to the Author**

1. Is the manuscript technically sound, and do the data support the conclusions?

Reviewer #1: Yes

Reviewer #2: Partly

2. Has the statistical analysis been performed appropriately and rigorously? 

Reviewer #1: Yes

Reviewer #2: Yes

3. Have the authors made all data underlying the findings in their manuscript fully available?

Reviewer #1: Yes

Reviewer #2: Yes

4. Is the manuscript presented in an intelligible fashion and written in standard English?

Reviewer #1: Yes

Reviewer #2: No

5. Review Comments to the Author

Reviewer #1: The study "Are total omega-3 and omega-6 polyunsaturated fatty acids predictors of fatal stroke in the Adventist Health Study 2 prospective cohort?" investigated the effects of n-3 PUFA, n-6 PUFA, and the ratio between the two on the risk of stroke mortality in a large prospective cohort (the Adventist Health Study-2 prospective cohort). The authors showed the subjects with greater total n-3 PUFA intakes had lower risk of fatal stroke, and those with a higher n-6 / n-3 PUFA ratio had higher risk.

Comments to Author:

Abstract: Please include the main objective of the study

Methods: Please provide detailed information about supplements intake.

Discussion: For discussion the 10th percentile (6.1:1) found in this study (percentile with lower risk of fatal stroke), suggest to the authors add a comment about the n-6/n-3 PUFA ratio found in studies with the Mediterranean Diet (diet known for the protection in cardiovascular and cerebrovascular events) or with the n-6/n-3 PUFA ratio found in longevity studies.

Reviewer #2: Dear Authors,

The manuscript has an interesting goal. However, the data needs to be reviewed.

Methods - What do the authors mean by this sentence, "The AHS-2 has been described in detail elsewhere..."? I suggest indicating proper where all the information has been described.

Methods- It is unclear how the authors investigated in the same category participants that had limited or no current consumption of tobacco and alcohol. It seems to me that they should be split into different groups.

Methods- I could not find the description appropriate to the acronym BMI. Even if it's a common acronym, it must be defined the first time it is included in the text.

Table 1 - What do the authors mean by the smoking ever used? Participants smoked in the past? For how long? How many cigarettes a day? For how long did they stop smoking? Please clarify.

-Dietary assessment- The authors chose poor methods of dietary assessment. It is a literature consensus that FFQ associated with 24 dietary recall is not appropriate to analyze food consumption. The more recent recommendation suggests using a 3-day food diary to explore the food consumption in the population.

- Dietary pattern- I could not understand if all participants are vegetarian or not. I suggest clarifying participants' dietary patterns.

- It is unclear which nutrients were supplemented or not. It made me a bit confused.

6. PLOS authors have the option to publish the peer review history of their article (what does this mean?). If published, this will include your full peer review and any attached files.

Reviewer #1: No

Reviewer #2: No

---

## [Author Response · Author response to Decision Letter 0]

2 Aug 2022

REVIEWER 1:

Abstract: Please include the main objective of the study

Thank you for your feedback. To highlight the objective of the study, we added the following new line to the abstract introduction: “This study therefore intended to investigate these effects in a cohort for which the vegetarian diet was more prevalent than the general public.”

Methods: Please provide detailed information about supplements intake.

To clarify the nature of the supplements, we added two new sentences to the Methods section under the Study Variables header: “Approximately 12% of participants supplemented n-3 PUFA, and approximately 15% supplemented n-6 PUFA. However, overall rates of supplementation were low: the 90th percentile of n-3 PUFA supplement contribution was 8.5% of total n-3 PUFA intake, and the corresponding 90th percentile of total n-6 PUFA intake due to n-6 PUFA supplements was 0.1%.”

Discussion: For discussion the 10th percentile (6.1:1) found in this study (percentile with lower risk of fatal stroke), suggest to the authors add a comment about the n-6/n-3 PUFA ratio found in studies with the Mediterranean Diet (diet known for the protection in cardiovascular and cerebrovascular events) or with the n-6/n-3 PUFA ratio found in longevity studies.

To improve our discussion about the n-6/n-3 PUFA ratio, we added a new sentence in the Discussion section to describe other populations’ n-6/n-3 PUFA ratios: “A previous review has investigated the n-6 / n-3 PUFA ratio of various diets and noted that while the present Western diet has an n-6 / n-3 PUFA ratio of 15-20:1, other populations have much lower ratios. Examples include Japan at 4:1 and rural India at 5-6:1, and specific diets such as Swedish at 4.7:1 or Mediterranean at 2.6:1 (52).”

REVIEWER 2:

Methods - What do the authors mean by this sentence, "The AHS-2 has been described in detail elsewhere..."? I suggest indicating proper where all the information has been described.

Thank you for this suggestion. We used this formal shorthand to denote that a separate journal article exists wherein the study is described in extensive detail. Recent PLOS One articles which have utilized similar statements include DOI=10.1371/journal.pone.0270033 (“Additional details are described elsewhere [18].”), DOI=10.1371/journal.pone.0269042 (“The detailed study design has been previously described [28].”), and DOI=10.1371/journal.pone.0270893 (“The study utilized MHS pregnancy registry, previously described [13].”). For improved clarity, we have updated the text to now read: “The AHS-2 has been described in detail previously by Butler et. al.”

Methods- It is unclear how the authors investigated in the same category participants that had limited or no current consumption of tobacco and alcohol. It seems to me that they should be split into different groups.

Historic tobacco and alcohol use were low in this cohort, so categories of use were collapsed. For clarity, we have added the following to a sentence in the Methods section describing tobacco and alcohol rates: “so variables controlling for those factors were collapsed”

Methods- I could not find the description appropriate to the acronym BMI. Even if it's a common acronym, it must be defined the first time it is included in the text.

The definition of BMI was added for the first use of the term, in the Methods section: “or body mass index (BMI).”

Table 1 - What do the authors mean by the smoking ever used? Participants smoked in the past? For how long? How many cigarettes a day? For how long did they stop smoking? Please clarify.

As previously mentioned, historic tobacco use was very low in this cohort so categories were collapsed. For clarification, the Methods section has now been altered to better define ever use: “current and/or prior use of tobacco; current and/or prior use of alcohol.”

-Dietary assessment- The authors chose poor methods of dietary assessment. It is a literature consensus that FFQ associated with 24 dietary recall is not appropriate to analyze food consumption. The more recent recommendation suggests using a 3-day food diary to explore the food consumption in the population.

Our review of the literature did not demonstrate a consensus supporting the statement “FFQ associated with 24 dietary recall is not appropriate to analyze food consumption.” However, we were able to find an article from the United States agency National Institutes of Health (DOI=10.1016/j.jand.2015.08.016) which states: “Even though there is bias in 24-hour recalls, because food records potentially have reactivity bias, the 24-hour recall is considered the least biased of the self-report instruments and the best single dietary assessment instrument for many purposes.” To that end, at least two validation papers have been published on the AHS-2 thus far, supporting the use of 24-hour recalls with a food frequency questionnaire: DOI=10.1017/S1368980009992072 and DOI=10.3945/jn.115.225508. Additionally, the AHS-2 has been conducted since 2002 and the authors of the present manuscript are unable to modify the protocols of the prospective cohort described in this manuscript.

- Dietary pattern- I could not understand if all participants are vegetarian or not. I suggest clarifying participants' dietary patterns.

We clarified by adding a new sentence in the Methods section to describe study participants’ dietary patterns: “Within this final analytical group, 37% were either vegan or lacto-ovo vegetarian, and 16% were either pesco-vegetarian or semi-vegetarian.”

- It is unclear which nutrients were supplemented or not. It made me a bit confused.

To improve clarity, we added a new sentence in the Methods section to list which nutrients were included from supplements: “Supplemented nutrients included saturated fatty acids, mono-unsaturated fatty acids, poly-unsaturated fatty acids, fiber, sodium, vitamin C, alpha-tocopherol equivalents, and flavonoids.”

---

## [Decision Letter · Decision Letter 1]

23 Aug 2022

Are total omega-3 and omega-6 polyunsaturated fatty acids predictors of fatal stroke in the Adventist Health Study 2 prospective cohort?

PONE-D-22-10872R1

Dear Dr. Cupino,

We’re pleased to inform you that your manuscript has been judged scientifically suitable for publication and will be formally accepted for publication once it meets all outstanding technical requirements.

Kind regards,

Alvaro Reischak-Oliveira, Ph.D.

Academic Editor

PLOS ONE

Reviewers' comments:

Reviewer's Responses to Questions

**Comments to the Author**

1. If the authors have adequately addressed your comments raised in a previous round of review and you feel that this manuscript is now acceptable for publication, you may indicate that here to bypass the “Comments to the Author” section, enter your conflict of interest statement in the “Confidential to Editor” section, and submit your "Accept" recommendation.

Reviewer #2: All comments have been addressed

2. Is the manuscript technically sound, and do the data support the conclusions?

Reviewer #2: Yes

3. Has the statistical analysis been performed appropriately and rigorously? 

Reviewer #2: N/A

4. Have the authors made all data underlying the findings in their manuscript fully available?

Reviewer #2: Yes

5. Is the manuscript presented in an intelligible fashion and written in standard English?

Reviewer #2: Yes

6. Review Comments to the Author

Reviewer #2: (No Response)

7. PLOS authors have the option to publish the peer review history of their article (what does this mean?). If published, this will include your full peer review and any attached files.

Reviewer #2: No

---

## [Editor Report · Acceptance letter]

30 Aug 2022

PONE-D-22-10872R1 

Are total omega-3 and omega-6 polyunsaturated fatty acids predictors of fatal stroke in the Adventist Health Study 2 prospective cohort? 

Dear Dr. Cupino:

I'm pleased to inform you that your manuscript has been deemed suitable for publication in PLOS ONE. Congratulations! Your manuscript is now with our production department. 

Kind regards, 

on behalf of

Dr. Alvaro Reischak-Oliveira 

Academic Editor

PLOS ONE